# Recording Sodium Self-Inhibition of Epithelial Sodium Channels Using Automated Electrophysiology in *Xenopus* Oocytes

**DOI:** 10.3390/membranes13050529

**Published:** 2023-05-19

**Authors:** Rene Y. Lawong, Fabian May, Etang C. Etang, Philipp Vorrat, Jonas George, Julia Weder, Dagmar Kockler, Matthias Preller, Mike Althaus

**Affiliations:** Department of Natural Sciences, Institute for Functional Gene Analytics, Bonn-Rhein-Sieg University of Applied Sciences, 53359 Rheinbach, Germany

**Keywords:** ENaC, sodium self-inhibition, hypertension, two-electrode voltage clamp, automated electrophysiology

## Abstract

The epithelial sodium channel (ENaC) is a key regulator of sodium homeostasis that contributes to blood pressure control. ENaC open probability is adjusted by extracellular sodium ions, a mechanism referred to as sodium self-inhibition (SSI). With a growing number of identified ENaC gene variants associated with hypertension, there is an increasing demand for medium- to high-throughput assays allowing the detection of alterations in ENaC activity and SSI. We evaluated a commercially available automated two-electrode voltage-clamp (TEVC) system that records transmembrane currents of ENaC-expressing *Xenopus* oocytes in 96-well microtiter plates. We employed guinea pig, human and *Xenopus laevis* ENaC orthologs that display specific magnitudes of SSI. While demonstrating some limitations over traditional TEVC systems with customized perfusion chambers, the automated TEVC system was able to detect the established SSI characteristics of the employed ENaC orthologs. We were able to confirm a reduced SSI in a gene variant, leading to C479R substitution in the human α-ENaC subunit that has been reported in Liddle syndrome. In conclusion, automated TEVC in *Xenopus* oocytes can detect SSI of ENaC orthologs and variants associated with hypertension. For precise mechanistic and kinetic analyses of SSI, optimization for faster solution exchange rates is recommended.

## 1. Introduction

The epithelial sodium channel (ENaC) is a sodium-selective ion channel in the apical membrane of the epithelial cells of the aldosterone-sensitive distal nephron, lung and colon, where it mediates the rate-limiting step for transepithelial sodium absorption. ENaC is a key component of the renin–angiotensin–aldosterone system, and hormonally controlled ENaC-activity matches dietary sodium intake to its excretion rate. ENaC is therefore a key protein in the control of sodium homeostasis. Since plasma sodium concentrations correlate with the extracellular fluid volume and systolic blood pressure [1], ENaC contributes to blood pressure control. Mutations in ENaC genes leading to increased channel activity cause Liddle syndrome, a hereditary form of hypertension [2,3].

A reduction in dietary sodium intake has been suggested to lower blood pressure; however, individual blood pressure correlations with dietary sodium intake vary. Some individuals respond strongly to an increase in dietary sodium intake with increased blood pressure, a condition described as salt-sensitive hypertension [4]. Recent genome analyses revealed that variants in ENaC-coding genes correlate with salt-sensitive blood pressure and might therefore predispose individuals to the development of hypertension [5,6,7]. Characterizing the functional consequences of ENaC gene variants is therefore an important goal towards the understanding of human predisposition to salt-sensitive hypertension.

ENaC activity is affected by the channel subunit assembly as well as intra- and extracellular stimuli controlling its membrane abundance and open probability (P_O_) [8]. There are four ENaC subunits, α-ENaC (*SCNN1A* gene), β-ENaC (*SCNN1B* gene), γ-ENaC (*SCNN1G* gene) and δ-ENaC (*SCNN1D* gene). Recent cryo-electron microscopy-derived structures revealed that each subunit resembles a clenched hand holding a central ball of beta-sheets [9,10]. The extracellular domains of each subunit are therefore termed ‘knuckle’, ‘finger’, ‘thumb’, ‘palm’ and ‘wrist’. Heterologous expression studies in *Xenopus* oocytes have shown that αβγ- and δβγ-subunit assemblies form functional ENaCs [11,12], with δβγ-ENaC consistently showing larger channel activity [13,14,15]. A key regulatory mechanism linking ENaC activity to extracellular sodium concentration is termed sodium self-inhibition (SSI). Extracellular sodium ions likely interact with an ‘acidic cleft’ located in the extracellular domains of the α-ENaC subunit [10,16] and trigger conformational changes that cause a reduction in P_O_ [16]. This mechanism has been suggested to prevent excess sodium absorption in the distal nephron under high extracellular sodium concentrations in the distal tubule [17]. Many additional intra- and extracellular regulatory pathways and stimuli elicit their effect on ENaC activity by altering the magnitude of SSI, thereby adjusting ENaC P_O_ [17]. SSI is therefore an important read-out for ENaC activity.

SSI was initially observed in transepithelial current recordings on frog skin preparations [18]. A rapid increase in the extracellular sodium concentration triggered a transient current increase that decreased within a few seconds to a steady state. SSI is defined as the decrease from the maximum current signals following high extracellular sodium concentrations to the steady state. Numerous studies also recorded SSI in ENaC expressing *Xenopus* oocytes using the two-electrode voltage-clamp (TEVC) technique (reviewed in [17]). Here, a rapid switch in the extracellular perfusion solution from low to high sodium concentrations triggers a similar transient increase in inward currents, followed by a decrease to a steady state due to SSI. *Xenopus* oocytes are therefore a valuable tool for studying SSI and investigating the functional effects of gene variants on SSI since mutated ENaCs can easily be expressed in these cells.

With an increasing number of identified ENaC gene variants, there is an increasing demand for medium- to high-throughput assays allowing the detection of alterations in ENaC SSI. We therefore aimed to establish a simple assay allowing us quantify SSI using a commercially available automated TEVC system recording transmembrane currents of ENaC expressing *Xenopus* oocytes in 96-well microtiter plates [19]. While the system has some limitations compared with traditional TEVC systems, we demonstrate that it can detect species-specific changes in SSI of ENaC orthologs and in a human α-ENaC gene variant (C479R) associated with Liddle syndrome.

## 2. Materials and Methods

### 2.1. Plasmids and cRNA Synthesis

Coding DNA sequences of human, guinea pig and *Xenopus laevis* ENaC subunits (α-, β-, γ- and δ-ENaC) were present in the pTNT vector (Promega, Walldorf, Germany). Plasmids were cleaned using the Monarch PCR and DNA Cleanup Kit (New England Biolabs, Frankfurt am Main, Germany) according to the manufacturer’s instructions, eluted in nuclease-free water and stored at −20 °C. Cleaned plasmids served as templates for cRNA synthesis using the mMESSAGE mMACHINE T7 Kit (Thermo Fisher Scientific, Waltham, MA, USA) in accordance with the manufacturer’s instructions. The cRNA was treated with TURBO DNase (New England Biolabs) to remove the template DNA, cleaned using the Monarch^®^ RNA Cleanup Kit (New England Biolabs) or MEGAclear Transcription Clean-Up Kit (Thermo Fisher Scientific) according to manufacturer’s instructions and eluted in nuclease-free water. The subunit cRNAs were combined as αβγ- or δβγ-ENaC combinations at a concentration of 5 ng/µL per ENaC subunit and stored at −80 °C.

### 2.2. Site-Directed Mutagenesis

Cysteine 479 in the human α-ENaC subunit was substituted by arginine using the Q5 Site-directed Mutagenesis Kit (New England Biolabs) according to manufacturer’s instructions. The forward primer sequence was 5’-CCGGAAGCCAcggAGCGTGACCA-3’, and the reverse primer sequence was 5’-CACTTGGTGAAACAGCCCAGG-3’. After DpnI digest of the template plasmid DNA, the mutated plasmids were heat-transformed into NEB 5-alpha Competent *E. coli* (New England Biolabs) and prepared using the Monarch^®^ Plasmid DNA Miniprep Kit (New England Biolabs) according to manufacturer’s instructions. Successful C479R substitution was confirmed using the sequencing of plasmid DNA, and the corresponding plasmids were employed as templates for cRNA synthesis as described in Section 2.1.

### 2.3. ENaC Expression in Xenopus laevis Oocytes

Stage V/VI *Xenopus laevis* oocytes were purchased from Ecocyte Bioscience (Dortmund, Germany) and stored in Modified Barth’s Solution (MBS; containing, in mM, 88 NaCl, 1 KCl, 2.4 NaHCO_3_, 0.82 MgSO_4_, 0.33 Ca(NO_3_)_2_, 0.41 CaCl_2_ and 10 HEPES (4-(2-hydroxyethyl)-1-piperazineethanesulfonic acid), pH = 7.5, and supplemented with 20 µg/mL gentamycin) at 4 °C until use. The MBS was exchanged after every 48 h. For injection of cRNA, the oocytes were placed into a *N*-Methyl-*D*-Glutamine Oocyte Ringer Solution (NMDG-ORS; containing (in mM) 10 NaCl, 1 KCl, 2 CaCl_2_, 2.5 Na^+^-pyruvate, 80 NMDG, 5 HEPES, pH = 7.4, and supplemented with 20 µg/mL gentamycin). The oocytes were manually injected with 18.4 nL of cRNA using a Nanoject II (Drummond Scientific Company, Broomall, PA, USA). Oocytes were stored in NMDG-ORS at 16 °C for 24 h.

### 2.4. Automated Two-Electrode Voltage-Clamp (TEVC) Electrophysiology

Automated TEVC recordings were performed using a Roboocyte 2 and corresponding measuring heads carrying the recording and bath electrodes (Multichannelsystems, Reutlingen, Germany). The capillaries of the recording electrodes were replaced by microelectrodes that were pulled from borosilicate glass capillaries (World Precision Instruments, Hitchin, UK; TW150-4) using a PP-83 (Narishige, Japan) or DMZ (Zeiss, München, Germany) puller. Recording electrodes were filled with 3 M KCl. Resistance of the current and voltage electrodes ranged between 100 and 2000 Ohm. Oocytes were placed in a 96-well microtiter plate containing 200 µL of low-sodium ORS (containing (in mM) 1 NaCl, 1 KCl, 2 CaCl_2_, 89 NMDG, 5 HEPES, pH = 7.4). Oocytes were clamped at a holding potential of −60 mV, and whole-cell transmembrane current signals (I_M_) were recorded at a sampling rate of 20 Hz. ENaC SSI was in the range of several seconds [20,21,22] and the relatively low sampling rate did therefore not interfere with detection of maximum SSI. Amplifier gain-P was 1000 nA/mV and amplifier gain-I was 100 1/s. Oocytes were superfused with low-sodium ORS or high-sodium ORS (containing (in mM) 90 NaCl, 1 KCl, 2 CaCl_2_, 5 HEPES, pH = 7.4) using the integrated Roboocyte 2 perfusion system. The perfusion rate yielding robust detection of SSI was tested in the initial experiments and eventually set to 6 mL/min (valve and waste pump speed: 7000) unless otherwise stated. Recordings were performed at buffer temperatures of 19 °C (room temperature) or 24 °C. To achieve 24 °C at the oocytes, the buffer reservoirs were placed in a water bath, and the outflow temperature at the measuring head was controlled. All recordings were performed using the Roboocyte 2 automated recording mode according to manufacturer’s instructions.

### 2.5. Chemicals and Reagents

All buffer ingredients were purchased from Carl Roth (Karlsruhe, Germany), except for NMDG, which was obtained from SIGMA, and sodium pyruvate and gentamycin which were obtained from Thermo Fisher Scientific (Waltham, MA, USA). The ENaC inhibitor amiloride (Tocris, Wiesbaden-Nordenstadt, Germany) was prepared as a 100 mM stock solution in water and employed in a final concentration of 100 μM to assure maximum block of both αβγ-ENaC and δβγ-ENaC isoforms.

### 2.6. Molecular Modelling

Mutation C479R was introduced in silico into the cryo-EM structure of the α subunit of the extracellular domain of αβγ-ENaC (PDB: 6WTH) [10] using the Schrödinger software suite (version 2023-01) [23,24]. The mutated structure was prepared and pre-processed with the Protein Preparation Wizard, and it was energy-minimized using Prime [25] and MacroModel [26] of the Schrödinger Suite and the OPLS4 force field [27]. The Polak–Ribiere conjugate gradient algorithm was used for energy minimization to a gradient of 10^−4^ kJ·mol^−1^·Å^−1^.

### 2.7. Data Analyses and Statistics

Data are presented as means ± standard deviation (SD). Experiments were performed using at least two independent *Xenopus* oocyte deliveries and the number of experimental repeats is given as *n*. Normal distribution of data was analyzed using the D’Agostino and Person test and corresponding parametric or non-parametric statistical analyses were performed as specified in the figure legends. All statistical analyses were performed using the GraphPad Prism version 9.5.1 (GraphPad Software, San Diego, CA, USA), and a *p*-value ≤ 0.05 was considered statistically significant.

## 3. Results

### 3.1. Evaluating Automated SSI Recordings Using Mammalian and Amphibian ENaC Orthologs

We initially recorded SSI in oocytes expressing guinea pig or *Xenopus laevis* αβγ- or δβγ-ENaCs, since these ENaC orthologs display characteristic magnitudes of SSI. We performed experiments at 19 °C and 24 °C since SSI is a temperature-dependent mechanism. To record SSI, ENaC-expressing oocytes were first perfused in a low-sodium ORS containing 1 mM NaCl for 30 s. The perfusion was subsequently switched to high-sodium ORS containing 90 mM NaCl for 3 min. Afterwards, the ENaC inhibitor amiloride (100 μM) was added to the high-sodium ORS to determine the baseline for ENaC-mediated inward currents. SSI was detectable in recordings from oocytes expressing guinea pig αβγ-ENaC (Figure 1). At 19 °C, SSI was 44.09 ± 20.47 % (*n* = 12), and it was 65.66 ± 10.95 % (*n* = 17) at 24 °C. Based on manual TEVC recordings, we have previously reported that guinea pig δβγ-ENaC has a strongly reduced SSI of approx. 16 % [15]. Using the automated TEVC system, SSI was mostly absent and determined as 4.24 ± 5.25 % (*n* = 9) at 19 °C, and −2.40 ± 8.36 % (*n* = 11) at 24 °C (Figure 1).

In contrast to guinea pig δβγ-ENaC, δβγ-ENaC from *Xenopus laevis* has a larger SSI than the corresponding αβγ-ENaC isoform [20]. Consistently, we observed high SSI in *Xenopus laevis* δβγ-ENaC (Figure 2) with 56.23 ± 9.18 % (*n* = 19) at 19 °C and 67.04 ± 11.96 % (*n* = 9) at 24 °C, whereas SSI of *Xenopus laevis* αβγ-ENaC was significantly lower with 10.57 ± 11.33 % (*n* = 11) at 19 °C, and 26.19 ± 2.8.09 % (*n* = 8) at 24 °C (Figure 2).

### 3.2. Recording SSI in a Human ENaC Mutation Associated with Liddle Syndrome

We next aimed to characterize SSI in human ENaC isoforms. Similar to guinea pig ENaC isoforms, a reduced SSI has been shown for human δβγ-ENaC compared with human αβγ-ENaC [15,21]. Furthermore, a mutation leading to C479R substitution in the α-ENaC subunit has been reported in Liddle syndrome [28]. To predict the consequence of the single-point mutation on the structure of the ENaC channel, we used in silico mutagenesis and molecular mechanics optimization (Figure 3). In the wildtype α-ENaC subunit, C479 forms a disulfide bridge with residue C394, which stabilizes the loop region between the β9 strand and the α4 helix (Figure 3b). This part connects the ‘palm’ and ‘thumb’ regions and is suggested to play a critical role during the conformational changes in the ENaC channel along the open-to-closed transition [10]. The disulfide bridge is broken in the mutated α-ENaC subunit (Figure 3c). In addition, the positive charge of the inserted arginine leads to the repulsion of surrounding residues with positively charged sidechains, and hence to a rearrangement of the entire loop region in our energy-minimized structure of mutated ENaC. Consequently, the reduced stability of the loop region might negatively affect intrinsic conformational changes in the ion channel and interactions with the nearby lipid membrane. Consistent with the importance of the region for the gating process in closely related acid-sensing ion channels (ASICs) which display a desensitization mechanism that has similarities with ENaC SSI, a reduced SSI for α_C479R_βγ-ENaC has been reported [29].

Using automated TEVC, we recorded SSI of human αβγ-ENaC of 39.24 ± 11.97 % (*n* = 18) at 19 °C and of 55.16 ± 9.36 % (*n* = 14) at 24 °C. SSI of human δβγ-ENaC was significantly lower (24.37 ± 9.21 % (*n* = 17) at 19 °C, and 5.37 ± 14.33 % (*n* = 11) at 24 °C) (Figure 4). Human α_C479R_βγ-ENaC also had a significantly reduced SSI of 11.50 ± 10.78 % (*n* = 17) at 19 °C and 25.74 ± 17.02 % (*n* = 13) at 24 °C. In summary, these data demonstrate that the automated TEVC on ENaC-expressing *Xenopus* oocytes in 96 well microtiter plates can be employed to investigate the effects of ENaC subunit gene variants on SSI.

## 4. Discussion

In this study, we aimed to record ENaC SSI in *Xenopus* oocytes using automated TEVC electrophysiology. SSI is a regulatory mechanism adjusting ENaC P_O_ to the extracellular sodium concentrations. A putative sodium binding site was identified in an ‘acidic cleft’ located between the ‘knuckle’ and ‘finger’ domains of the extracellular region in the mammalian α-ENaC [10,16] and *Xenopus* δ-ENaC subunits [30]. It is suggested that the binding of sodium ions to this region triggers conformational changes in the extracellular domains that are linked to ENaC gating and maintain the channel in a low P_O_ state [10,16,30]. Alteration in ENaC activity due to various stimuli is often associated with changes in the magnitude of SSI, with a release from SSI activating the channel, and enhancement of SSI causing channel inhibition [30]. SSI is therefore a suitable read out for ENaC activity, and it is important to investigate the effect of ENaC subunit gene variants on SSI.

Recording of SSI requires the separation of sodium-mediated ion currents from the inhibition of ion currents due to sodium binding to ENaC. We have previously recorded SSI in ENaC-expressing oocytes using traditional TEVC systems [15,30]. In these studies, the transient SSI peak currents were detected by rapidly switching the extracellular perfusion solutions from low (1 mM) to high (90 mM) sodium concentrations using high perfusion speeds (10–12 mL/min) and custom-made, laminar perfusion chambers allowing optimal fluid exchange. The herein-employed automated TEVC system operates in 96-well microtiter plates and does not achieve such high perfusion speeds. Furthermore, the input and output tubing of the perfusion system are both placed closely together above the oocytes, suggesting that turbulent fluid flow during application of different solutions is likely to occur. Nevertheless, we were able to record SSI in guinea pig and *Xenopus laevis* ENaC orthologs and confirmed, consistent with previous publications [15,20], the reduced (guinea pig) or enhanced (*Xenopus laevis*) SSI in δβγ-ENaCs compared with the corresponding αβγ-ENaCs. However, the herein-employed system shows limitations over traditional TEVC systems with optimized perfusion chambers, since SSI in guinea pig δβγ-ENaCs was almost absent in the automated TEVC recordings, whereas 16% SSI could previously be recorded using traditional TEVC systems [15]. To exclude the potential effects of buffer temperatures on SSI magnitudes, we determined SSI at 19 °C and 24 °C. While magnitudes of SSI were generally larger at 24 °C, SSI in guinea pig δβγ-ENaCs was still largely absent. This is most likely due to a reduced rate of sodium exchange in the perfusion buffers in the 96-well plates compared with perfusion chambers. Nevertheless, it was possible to detect specific changes in SSI using automated TEVC recordings.

We therefore aimed to record changes in SSI in a human α-ENaC variant (C479R) that was identified in a family with Liddle syndrome [28]. Salih et al. reported that α_C479R_βγ-ENaC displayed larger currents than wild type αβγ-ENaC when expressed in *Xenopus* oocytes [28]. The authors did not detect any changes in membrane abundance and reported a reduced sensitivity to the ENaC-activating protease trypsin, suggesting a reduced P_O_ of α_C479R_βγ-ENaC. Furthermore, a reduction in SSI was described for this ENaC variant [29]. Molecular modeling revealed that substitution of cysteine 479 by arginine disrupts a disulfide bond and destabilizes the loop region between the β9 strand of the ‘palm’ and the α4 helix of the ‘thumb’ domains. The ‘palm’, ’thumb’ and ‘wrist’ domains have previously been shown to play a role in ENaC gating and SSI [22,31,32,33]. However, the precise structural mechanisms of ENaC gating are unknown due to low resolution of the available cryo-electron microscopy-derived structures in the transmembrane and pore region [9]. However, cysteine 479 is highly conserved in the degenerin/ENaC protein family [9] and structural information of homologous chicken ASIC1a in an open and closed conformation revealed that movement of the ‘thumb’ domain and the β9-α4 loop near the ‘wrist’ domain is associated with channel gating [34]. In agreement with the previous reports [15,21], we detected a reduced SSI in human δβγ-ENaCs compared with αβγ-ENaCs. We were able to confirm a reduced SSI in human α_C479R_βγ-ENaC [29], suggesting a role of the β9-α4 loop in ENaC gating. Furthermore, lower SSI in human α_C479R_βγ-ENaC is consistent with a reduced sensitivity to trypsin [28] since proteases activate αβγ-ENaCs by decreasing SSI [15,35]. Notably, human α_C479R_βγ-ENaC yielded low transmembrane currents in our experiments; however, our experiments were not designed to compare overall transmembrane current signals, and SSI is independent of ENaC expression levels. Reduced SSI hampers ENaC from decreasing its P_O_ under conditions of high extracellular sodium concentrations. Consequently, enhanced sodium retention in the distal nephron provides a mechanistic explanation for hypertension in patients carrying this ENaC variant.

## 5. Conclusions

In conclusion, automated TEVC in *Xenopus* oocytes can reveal alterations in SSI of ENaC orthologs and variants associated with hypertension. For precise mechanistic and kinetic analyses of SSI, optimization for faster solution exchange rates is recommended.

## Figures and Tables

**Figure 1 membranes-13-00529-f001:**
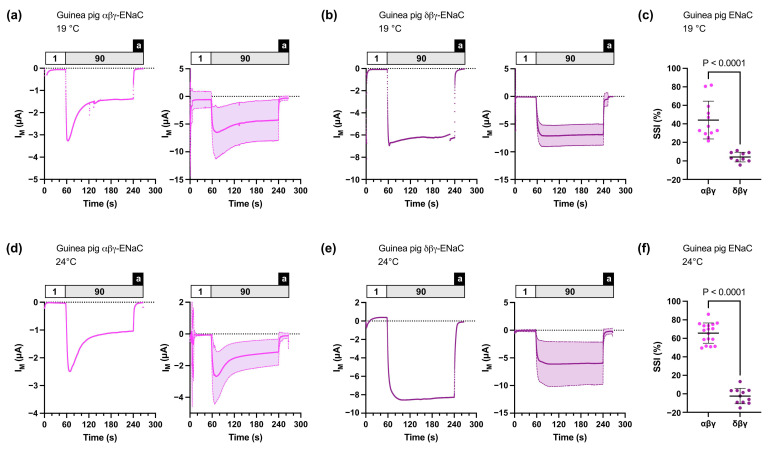
Sodium self-inhibition (SSI) of guinea pig αβγ- or δβγ-ENaCs heterologously expressed in *Xenopus* oocytes. (**a**) Left panel: representative transmembrane (I_M_) recording of an oocyte expressing αβγ-ENaC at 19 °C; right panel: mean ± SD plotted from experiments as shown in the left panel (*n* = 12). (**b**) Left panel: representative transmembrane (I_M_) recording of an oocyte expressing δβγ-ENaC at 19 °C; right panel: mean ± SD plotted from experiments as shown in the left panel (*n* = 9). (**c**) SSI was calculated as the percent decrease in amiloride-baseline subtracted I_M_ within 3 min after switching the extracellular solution from low-sodium (1) to high-sodium (90) ORS. For αβγ-ENaC, *n* = 12; for δβγ-ENaC, *n* = 9. Student’s unpaired t-test (two-tailed). (**d**,**e**) Similar to panels (**a**,**b**) with recordings performed at 24 °C, for αβγ-ENaC, *n* = 17; for δβγ-ENaC, *n* = 11. (**f**) Similar to panel (**c**) with data from recordings performed at 24 °C, for αβγ-ENaC, *n* = 17; for δβγ-ENaC, *n* = 11. Student’s unpaired *t*-test (two tailed). Notes: 1 = low-sodium ORS containing 1 mM NaCl; 90 = high-sodium ORS containing 90 mM NaCl; a = amiloride.

**Figure 2 membranes-13-00529-f002:**
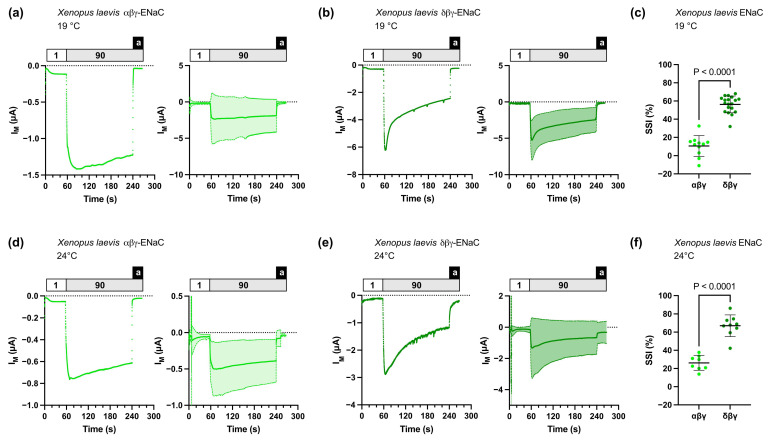
Sodium self-inhibition (SSI) of *Xenopus laevis* αβγ- or δβγ-ENaCs heterologously expressed in *Xenopus* oocytes. (**a**) Left panel: representative transmembrane (I_M_) recording of an oocyte expressing αβγ-ENaC at 19 °C; right panel: mean ± SD plotted from experiments as shown in the left panel (*n* = 11). (**b**) Left panel: representative transmembrane (I_M_) recording of an oocyte expressing δβγ-ENaC at 19 °C; right panel: mean ± SD plotted from experiments as shown in the left panel (*n* = 19). (**c**) SSI was calculated as the percent decrease in amiloride-baseline subtracted I_M_ within 3 min after switching the extracellular solution from low-sodium (1) to high-sodium (90) ORS. For αβγ-ENaC, *n* = 11; for δβγ-ENaC, *n* = 19. Student’s unpaired t-test (two-tailed). (**d**,**e**) Similar to panels (**a**,**b**) with recordings performed at 24 °C, for αβγ-ENaC, *n* = 8; for δβγ-ENaC, *n* = 9. (**f**) Similar to panel (**c**) with data from recordings performed at 24 °C, for αβγ-ENaC, *n* = 8; for δβγ-ENaC, *n* = 9. Student’s unpaired *t*-test (two-tailed). Notes: 1 = low-sodium ORS containing 1 mM NaCl; 90 = high-sodium ORS containing 90 mM NaCl; a = amiloride.

**Figure 3 membranes-13-00529-f003:**
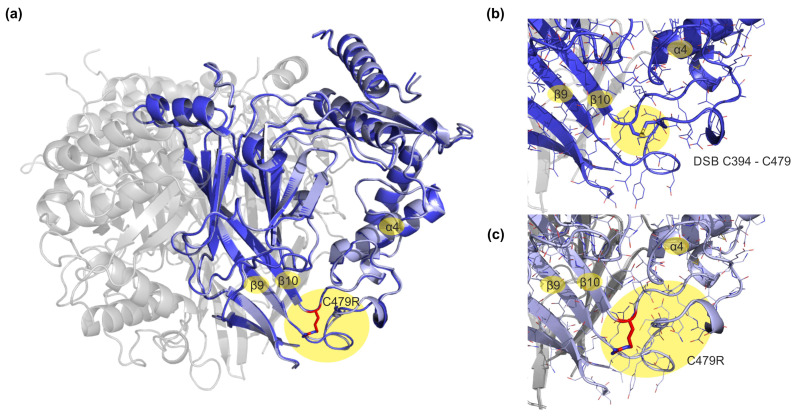
Structural model of the human α-ENaC subunit after energy minimization. (**a**) Superposition of the energy-minimized wildtype α-ENaC subunit (dark blue cartoon representation) and the C479R mutation (light blue cartoon and red sticks representation) indicates local rearrangements of the loop region between beta-strand β9 and the α4 helix. This region is suggested as a fulcrum for conformational changes during the open–closed transition of the ENaC channel. Subunits β and γ are shown in grey. (**b**) Close-up view on the wildtype α subunit of ENaC (dark blue) with the conserved disulfide bridge between C394 and C479. This disulfide bridge appears to stabilize the adjacent loop region. (**c**) Close-up view on the mutated α-ENaC subunit (light blue) shows that the inserted, positively charged arginine at position 479 leads to the repulsion of surrounding, positively charged amino acids of the loop, thereby potentially affecting intrinsic conformational changes and interactions with the lipid membrane.

**Figure 4 membranes-13-00529-f004:**
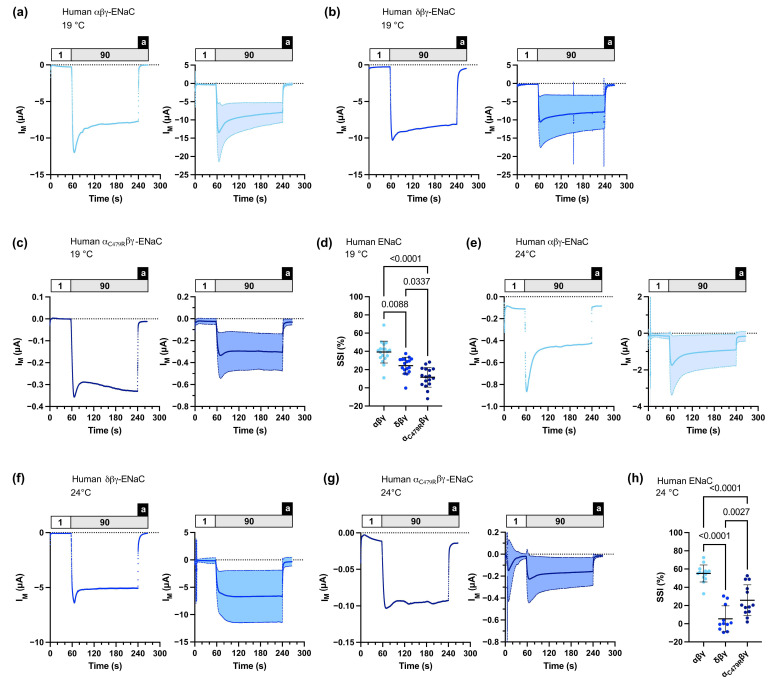
Sodium self-inhibition (SSI) of human αβγ-, δβγ-, or α_C479R_βγ-ENaCs heterologously expressed in *Xenopus* oocytes. (**a**) Left panel: representative transmembrane (I_M_) recording of an oocyte expressing αβγ-ENaC at 19 °C; right panel: mean ± SD plotted from experiments as shown in the left panel (*n* = 18). (**b**) Left panel: representative transmembrane (I_M_) recording of an oocyte expressing δβγ-ENaC at 19 °C; right panel: mean ± SD plotted from experiments as shown in the left panel (*n* = 17). (**c**) Left panel: representative transmembrane (I_M_) recording of an oocyte expressing α_C479R_βγ-ENaC at 19 °C; right panel: mean ± SD plotted from experiments as shown in the left panel (*n* = 17). (**d**) SSI was calculated as the percent decrease in amiloride-baseline subtracted I_M_ within 3 min after switching the extracellular solution from low-sodium (1) to high-sodium (90) ORS. For αβγ-ENaC, *n* = 18; for δβγ-ENaC, *n* = 17; for α_C479R_βγ-ENaC, *n* = 17. Kruskal–Wallis test followed by Dunn’s multiple comparisons test. (**e**–**g**) Similar to panels (**a**–**c**) with recordings performed at 24 °C, for αβγ-ENaC, *n* = 14; for δβγ-ENaC, *n* = 11; for α_C479R_βγ-ENaC, *n* = 13. (**h**) Similar to panel (**d**) with data from recordings performed at 24 °C, for αβγ-ENaC, *n* = 14; for δβγ-ENaC, *n* = 11; for α_C479R_βγ-ENaC, *n* = 13. One-way ANOVA followed by Tukey’s multiple comparisons test. Notes: 1 = low-sodium ORS containing 1 mM NaCl; 90 = high-sodium ORS containing 90 mM NaCl; a = amiloride.

## Data Availability

Not applicable.

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
