# Peer review of "Recording Sodium Self-Inhibition of Epithelial Sodium Channels Using Automated Electrophysiology in Xenopus Oocytes"

_membranes, 2023, doi:10.3390/membranes13050529_

Round 1

Reviewer 1 Report

The manuscript is well-written and the research well organized. A figure with similar data collected by a classical two-electrode Voltage clamp can help to understand the differences between the automatic and classical systems.

The main problem of this ms is the resolution of the figures - It was really hard to read the subunits' composition and look carefully at the current dimensions. Even the figure with the structure is not at the appropriate resolution.

When a protein is the same protein but from a different organism it is better to define them as an ortholog and not as an isoform- I suggest correcting the text according.

the quality of English Language  is adequate

Author Response

Reviewer 1

Comment 1: The manuscript is well-written and the research well organized. A figure with similar data collected by a classical two-electrode Voltage clamp can help to understand the differences between the automatic and classical systems.

Response 1: We would like to thank the reviewer for their positive feedback on our manuscript. Unfortunately, as a result of the flash floods in Germany in July 2021, which severely impacted our campus and laboratories, we do not have the classical two-electrode votlage-clamp setup available in our laboratory at the moment. However, we recently published data which we obtained with the classical setup under the same conditions (see Figure 5 E in Gettings SM, Mol Biol Evol. 2021 9;38(12):5704-5725. doi: 10.1093/molbev/msab271.) and refer to these data in the manuscript. In order to avoid double-publication of this data we would prefer referencing this study (see lines 175/176 and 310-312) rather than reproducing the transmembrane current trace.

Comment 2: The main problem of this ms is the resolution of the figures - It was really hard to read the subunits' composition and look carefully at the current dimensions. Even the figure with the structure is not at the appropriate resolution.

Response 2: Many thanks for bringing this to our attention. The figures had a high resolution in our files and after investigation with the editorial office it turned out that the resolution was apparently altered during file processing. We apologize for this inconvenience. The revised manuscript should contain the figures in adequate quality.

Comment 3: When a protein is the same protein but from a different organism it is better to define them as an ortholog and not as an isoform- I suggest correcting the text according.

Response 3: Many thanks for this comment. In the manuscript we use both ENaC orthologs, but also ENaC isoforms (based on ENaC subunit composition). We corrected and clarified this accordingly throughout the manuscript.

Reviewer 2 Report

The authors tested the sodium self-inhibition (SSI) of epithelial sodium channels (ENaC) on Xenopus oocytes in an automated two-electrode voltage-clamp (TEVC) setup. Three different ENaC isoforms (guinea pig, human, and Xenopus laevis) were used as they have different SSIs.

Compared to the traditional TEVC setup, automated TEVC could record SSIs with some limited differences. Moreover, they tested the SSI in a human mutant ENaC channel associated with Liddle syndrome. The SSI of the mutant channel is similar between traditional and automated TEVC setups. They concluded that automated TEVC could record the SSI of ENaC isoforms. The manuscript is well-written. The date seems convincing, and the conclusion seems appropriate. Below are some minor concerns.

1. All figures are blurry.

2. Figure 1f, why are there some negative SSI data points? This issue also happens in Figure 4c. 

3. Line 231, do acid-sensing ion channels also show SSI? More explanation is needed here.

4. How to keep the experiment at 19 degrees and  24 degrees is missing in the method. 

English is fine. 

Author Response

Comment 1: The authors tested the sodium self-inhibition (SSI) of epithelial sodium channels (ENaC) on Xenopus oocytes in an automated two-electrode voltage-clamp (TEVC) setup. Three different ENaC isoforms (guinea pig, human, and Xenopus laevis) were used as they have different SSIs. Compared to the traditional TEVC setup, automated TEVC could record SSIs with some limited differences. Moreover, they tested the SSI in a human mutant ENaC channel associated with Liddle syndrome. The SSI of the mutant channel is similar between traditional and automated TEVC setups. They concluded that automated TEVC could record the SSI of ENaC isoforms. The manuscript is well-written. The date seems convincing, and the conclusion seems appropriate. Below are some minor concerns.

Response 1: We would like to thank the reviewer for their positive feedback on our manuscript.  

Comment 2: All figures are blurry.

Response 2: Many thanks for bringing this to our attention. The figures had a high resolution and after investigation with the editorial office it turned out that the resolution was apparently altered during file processing. We apologize for this inconvenience. The revised manuscript should contain the figures in adequate quality.

Comment 3: Figure 1f, why are there some negative SSI data points? This issue also happens in Figure 4c.

Response 3: The SSI is calculated as the % change in amiloride-sensitive currents before and after 3 min of exposure to 90 mM sodium containing solution. In cases where SSI is strongly reduced or absent, we sometimes observe a slight increase in amiloride-sensitive currents. This explains the negative SSI data.

Comment 4: Line 231, do acid-sensing ion channels also show SSI? More explanation is needed here.

Response 4: Acid-sensing ion channels (ASICs) do not demonstrate a sodium self-inhibition but display a (proton-induced) desensitization mechanism that is considered similar to ENaC SSI (Hanukoglu I. FEBS J. 2017 284(4):525-545. doi: 10.1111/febs.13840). ASICs and ENaCs are therefore considered to have similar gating mechanisms. We have added a brief statement on this in lines 234/235.

Comment 5: How to keep the experiment at 19 degrees and 24 degrees is missing in the method.

Response 5: 19 degrees was the room temperature in the lab and all buffers were equilibrated to this temperature (which we confirmed by measurements). To achieve 24 °C at the oocytes, we placed the buffers in a water bath and measured the outflow temperature at the measuring hat. We included this description in the methods section. We also added additional details on the settings for the automated recordings.

Round 2

Reviewer 2 Report

No comments. 

No comments.